# Weak Polyelectrolytes as Nanoarchitectonic Design Tools for Functional Materials: A Review of Recent Achievements

**DOI:** 10.3390/molecules27103263

**Published:** 2022-05-19

**Authors:** Noelia M. Sanchez-Ballester, Flavien Sciortino, Sajjad Husain Mir, Gaulthier Rydzek

**Affiliations:** 1ICGM, CNRS, ENSCM, University of Montpellier, 34000 Montpellier, France; noelia.sanchez-ballester@umontpellier.fr; 2Department of Pharmacy, Nîmes University Hospital, 30029 Nîmes, France; 3Department of Chemistry, University of Basel, Mattenstrasse 24a, 4002 Basel, Switzerland; flavien.sciortino@unibas.ch; 4School of Chemistry and Advanced Materials & BioEngineering Research (AMBER) Center, Trinity College Dublin, The University of Dublin, D02 PN40 Dublin, Ireland

**Keywords:** polymer materials, polyelectrolyte films, electrodeposition, block copolymer, self-assembly, LbL, nanostructured materials, pharmaceutical vectors

## Abstract

The ionization degree, charge density, and conformation of weak polyelectrolytes can be adjusted through adjusting the pH and ionic strength stimuli. Such polymers thus offer a range of reversible interactions, including electrostatic complexation, H-bonding, and hydrophobic interactions, which position weak polyelectrolytes as key nano-units for the design of dynamic systems with precise structures, compositions, and responses to stimuli. The purpose of this review article is to discuss recent examples of nanoarchitectonic systems and applications that use weak polyelectrolytes as smart components. Surface platforms (electrodeposited films, brushes), multilayers (coatings and capsules), processed polyelectrolyte complexes (gels and membranes), and pharmaceutical vectors from both synthetic or natural-type weak polyelectrolytes are discussed. Finally, the increasing significance of block copolymers with weak polyion blocks is discussed with respect to the design of nanovectors by micellization and film/membrane nanopatterning via phase separation.

## 1. Context

Functional materials with precisely controlled compositions and (nano)structures and predictable responses to stimuli are prime candidates to address a number of challenges in fields including biomaterials, vectorization, and energy storage and conversion [1]. The development of such “smart” materials relies largely on designing functional systems through the assembly of well-controlled nanoscale units. In this context, the paradigm of “nanoarchitectonics” was proposed by Aono in the early 2000s [2] and subsequently developed by Ariga to enable materials that benefit from *(i)* the emergence of system-scale functions from the collective integration of nano-units (nanosystem functionality), *(ii)* robust function despite nano-unit/assembly defects (unreliability-tolerant reliability), and *(iii)* the emergence of unexpected new functions from large combinatorial systems (quantity changes quality). Thus, implementing this concept by combining tools from research fields including nanotechnology, (bio)material science, and organic and polymer chemistry has ensured that considerable advances have been made in the elaboration of dynamic materials and systems relying on reversible non-covalent interactions [3]. Typically, the reversible nature of interactions with and within functional nano-units provides materials with self-assembly synthesis routes, structural control, encapsulation abilities, and responses to stimuli [4]. In this context, polyelectrolytes occupy a central position in many nanoarchitectured systems owing to their multipotent properties, enabling them to act as reticulation/bridging agents, stabilizers, structure directors, and matrix components in functional coatings, gels, and colloidal systems [4,5].

Polyelectrolytes whose charge does not change over the entire pH range (1–14) are considered “strong“, while the density of charge on “weak” polyelectrolytes depends on the pH. In such a case, weak polyanions (resp. polycations) bear weak acid (resp. basic) moieties and their pKa value corresponds to the pH at which half of their repeating monomer units are charged. Consequently, the ionization degree of weak polyelectrolytes is intrinsically pH-responsive, with consequences for their conformation (such as extended chains being favored at high ionization degrees), H-bonding (being favored at low ionization degrees), and electrostatic complexation capabilities with other charged species (Figure 1). In addition, the apparent charge density of weak polyelectrolytes can be modulated through ion screening by changing the concentration of monovalent salts in solutions. This phenomenon, which represents a second type of response to stimuli, disfavors the electrostatic complexation of polyelectrolytes at high ionic strengths and changes their conformation, with swollen loops being favored at high ionic strengths [6]. Finally, when the valency of counter-ions is changed from monovalent to multivalent, intramolecular electrostatic crosslinking leads to further conformational transitions (Figure 1) [7,8]. It is thus not surprising that a wide range of synthetic and natural weak polyelectrolytes have been developed (Table 1), while their properties as reversible polyacids/bases [9,10], complexing agents [11,12], and self-assembling units in functional systems [13,14,15,16] have been very actively investigated. It is worth noting that the apparent pKa of weak polyelectrolytes, measured by pH titration, conductimetry, and infrared spectroscopy, varies notably depending on experimental parameters such as their molecular weight, conformation, confinement, and complexation [17,18,19,20].

Based on these backgrounds, weak polyelectrolytes offer a range of non-covalent interactions, including electrostatic, H-bonding, and hydrophobic interactions. On the one hand, these properties can not only ensure the cohesion of polymeric matrices, films, and colloids but also direct their structuration at the nano and micron levels (Figure 1). On the other hand, they offer routes to tailor the functions—including sorption, swelling, and mechanical responses—of systems whose cohesion relies on covalent bonds. The objective of this review article is to present recent examples of stimuli-responsive systems with triggerable interactions at the molecular and supramolecular levels by modulating the charge density on weak polyelectrolytes through salt and pH stimuli. The examples described cannot cover all aspects but rather focus on the most recent developments, generally reported less than 5 years ago. Accordingly, this review is organized into four sections, each dealing with a different nature of weak polyelectrolyte systems:-In the initial part, organic and hybrid thin films obtained from weak polyelectrolyte brushes and electrodeposited coatings are described, including those based on natural polyelectrolytes (e.g., chitosan, alginate, hyaluronic acid). The cohesion of such systems relies on covalent, hydrophobic and H-bonding interactions, most often leading to coatings containing one weak polyelectrolyte at a time. Basic synthesis approaches for these films are discussed as well as their applications, with an emphasis on their response to post-assembly pH changes in terms of swelling, adhesion, and cargo encapsulation and release.-The next two sections describe systems based on polyelectrolyte complexes, including those formed from the electrostatic complexation of weak polyanions and polycations [34]. Their assembly in aqueous media, without aggressive chemicals, allows for the use of more environmentally friendly routes to obtain surface coatings (films and layer-by-layer capsules), vectors, and functional gels. The resulting systems usually include several polyelectrolytes at the same time, and their cohesion is based on reversible interactions. The fundamentals of these films, gels, and colloids are discussed with respect to their response to pH and salt stimuli both during and after assembly. Emphasis is placed on recently proposed processing strategies to transform electrostatic complexes into gels and membranes. The applications of these systems are reviewed with a focus on nanovectors, and a subsection is devoted to systems that have been identified as relevant to pharmaceutical needs.-The final part addresses the growing significance of block copolymers (BCP) containing weak polyelectrolyte blocks for nanostructuring surfaces, colloids, and membranes. Accordingly, their directed self-assembly into microphase-separated films and their micellization behavior are discussed as a function of complexation, pH, and salt stimuli. Emerging applications, including sensors, nanolithography, and vectorization, are discussed.

**Figure 1 molecules-27-03263-f001:**
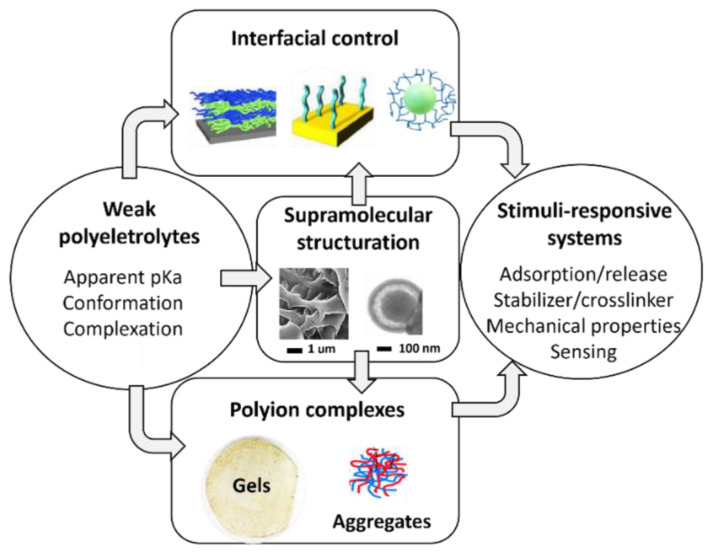
**Weak polyelectrolyte as a multipotent tool for stimuli-responsive systems.** Depiction of the different usages of weak polyelectrolytes for supporting the nanoarchitectonic design of functional materials addressing stimuli-responsive applications.

## 2. Weak Polyelectrolytes Layers for Stimuli-Responsive Surfaces

Coatings based on weak polyelectrolytes provide researchers with a convenient way to tune the properties of surfaces, ultimately adjusting their interactions through their environment and enabling the design of stimuli-responsive systems. This section aims to review brushes and electrodeposited coatings, most often composed of only one weak polyelectrolyte, as well as their applications with respect to their response to pH changes during and after assembly.

### 2.1. Brushes of Weak Polyelectrolytes

Polymer brushes are obtained through the chemical grafting of surfaces with a layer of polymers. The resulting coatings are considered more stable than those obtained from casting owing to their capability to withstand immersion in good solvent without dissolution. Polymer brushes are extensively used as colloidal stabilizers, lubricating layers, and drug delivery systems [35]. In that respect, elaborating brushes based on weak polyelectrolytes is generally aimed at developing stimuli-responsive systems, and was reviewed recently [15]. Grafting-from approaches, where polyelectrolyte chains grow in situ from their surface anchoring points, typically offering higher grafting densities than approaches where pre-synthesized polyelectrolyte chains are used [36,37], have therefore been developed by using several radical polymerization technics [35]. Accordingly, both polyanion [38] and polycation [39] brushes have been synthesized by using nitroxide-mediated polymerization (NMP). A larger variety of systems were made accessible using atom-transfer-radical-polymerization (ATRP) at the cost of using a metal catalyst, yielding routinely weak polycationic brushes such as PDMAEMA [40]. However, the growth of carboxylate-containing polyanions (PAA, PMAA) brushes is more complicated owing to interactions with the catalyst, leading researchers to polymerize their respective ester derivatives, followed by a deprotection step [41]. Finally, reversible addition-fragmentation chain transfer (RAFT), a catalyst-free polymerization method, was used for developing weak polycationic brushes [42,43] with the trade-off of using quite expensive chain transfer agents. However, this approach has so far been neglected with regard to the synthesis of weak polyanion brushes.

The pH response of weak polyelectrolyte brushes leads to conformational changes in polymer chains, resulting in thickness variations in the brush: when the ionization degree of the polyelectrolyte decreases (resp. increases), the brush tends to collapse (resp. swell). This effect is largely modulated by the concentration [44], type [45], and valency [7] of counter-ions and salt in the system. Changes in pH and ionic strength parameters can also be induced electrochemically, yielding a reversible swelling of brushes [46]. Yet, when designing such pH-responsive systems, one must keep in mind that the apparent pKa (or pKaH) of weak polyelectrolytes in brushes greatly varies with the intrinsic parameters, such as their grafting density [47] and ionic strength [48]. These responses have been exploited to design a range of functional systems, as described in Table 2. Among the most studied applications of polyelectrolyte brushes, both their lubricating and adhesion functions can be efficiently switched on and off by using weak polyelectrolytes [7,49]. Lubrication is ensured by the swollen hydrated state of polyelectrolyte brushes. Although polyzwitterionic brushes have been determined to perform better, brushes composed of weak polyelectrolytes offer an opportunity to modulate lubrication through pH and ionic (concentration, valency) stimuli [7]. Adhesion in water dependent on pH has also been demonstrated between a range of weak polyelectrolyte brushes, including PAA with poly(N,N-dimethylacrylamide) and PMAA with PDMAEMA. The adhesion mechanism is, however, different with H-bonding, ensuring the cohesion of the first polyelectrolyte couple (pH values smaller than 2) [50], while electrostatic interactions dominate for the PMAA/PDMAEMA couple (adhesion for pH values close to 7) [51]. The combination of electrostatic, H-bonding, and hydrophobic interactions in weak polyelectrolyte brushes enable cargo (e.g., proteins) immobilization with a large binding capacity, reversible nature, and structure-preserving ability [52,53]. However, in a recent example, Ferrand-Drake del Castillo et al. suggested that the dominant force for protein loading and release when using weak polyelectrolyte brushes was *not* electrostatic, owing to the large changes in their apparent pKa [48].

### 2.2. Electrodeposited Weak Polyelectrolytes Films

Electrochemically induced film deposition strategies have recently attracted a great deal of attention, as the localized nature of the electrochemical trigger enables spatially resolved film assembly in a conformal manner, including on substrates with complex topologies. Applications of such films are anticipated in many fields, as testified by pioneering work in biomaterials, energy storage and conversion, mass transport, and analytic tools [59]. Several electrochemical approaches to weak polyelectrolyte films have thus been developed by using electrochemically induced precipitation, electropolymerization, and electrochemically induced coupling reactions (Figure 2). In this section, these different strategies will be described, with a focus on experimental synthesis parameters and applications.

The electrochemically induced precipitation of weak polyelectrolyte films is typically achieved by locally changing the solubility of weak polyelectrolytes near the electrode, favoring their self-association and precipitation. This process can be triggered, among the most popular strategies, by changing the local pH through water electrolysis *(i)* or by enabling the complexation of the polyelectrolytes with multivalent ions *(ii)*. In this context, electrodeposited weak polyelectrolytes films have mainly been developed from pure and composite assemblies of CHI, ALG, PAH, and gelatin on a large range of electrodes (including patterned electrodes) with spatial and temporal control [63].

*(i)* Water electrolysis allows increasing the local pH value at the cathode, favoring the deposition of films based on weak polycations, such as CHI, through decreasing their ionization degree [63]. Here, again, the ionic strength of the solution during the film deposition directly influences the thickness and mechanical properties of the polyelectrolyte assembly, with thicker and softer films obtained at higher salt concentrations [64]. At the anode, the proton gradient generated during water electrolysis promotes film assembly from weak polyanions such as HA, ALG, and silk fibroin through protonation and self-association (Table 3). In contrast, a few examples have reported electrodeposition strategies where an increase in the charge density of weak polyelectrolytes was sought. In that case, proton and hydroxide ion gradients generated by electrolysis in aqueous solutions induce higher ionization degrees and/or acid deprotection of the weak polyelectrolytes, enabling their complexation and film precipitation with oppositely charged polymers [65,66,67] and multivalent MoO_4_^2−^ ions [68].

*(ii)* The complexation of weak polyelectrolytes with multivalent ions leading to their film precipitation is another prominent electrodeposition strategy. The generation of the crosslinking ions is achieved either by the dissolution of oxide precursors [69] or by direct oxidation to the relevant cations [70]. Anodic electrolysis has been used to dissolve CaCO_3_ (respectively, Cu_2_(OH)_2_CO_3_), yielding Ca^2+^ and Cu^2+^ ions near the electrode and allowing the assembly of ALG films by complexation [69,71]. Allowing work to be carried out at milder oxidation potentials, the electrochemical generation of multivalent cations has been developed with Cu^2+^, Fe^3+^, and Ru^2+^ cations, allowing complexation and film deposition with ALG, Chitin, CHI, and PAA modified with terpyridine groups [71,72,73].

In situ polymerization on electrodes by electrochemical reaction (electropolymerization) is another popular strategy that can ensure the elaboration of functional films. Aniline, dopamine, carbazole, and vinylpyridine monomers and their derivatives have been extensively studied for use in the electropolymerization of weak polyelectrolyte films under oxidative potentials (Table 3). The resulting films can combine electron conductivity with weak polybase characteristics, conferring them with doping, complexation, and molecular imprinting abilities. A large number of recent studies have used these appealing properties to design sensors, energy devices, light-emitting devices, and electrochromic and “smart” electrodes, as described in recent reviews [72,73,74,75,76]. The presence of aromatic cycles in such polymers also confers them with a hydrophobic character and self-assembling properties that enable them not only to act as weak polyelectrolytes but also as structure directing nano-units (Figure 2B). For instance, PANI self-organization has been exploited to design nanostructured colloids [77] and coatings [61,78], which are optimized as catalytic supports and energy storage platforms, respectively. Accordingly, several self-organized structured polyvinylpyrridine sensors have been reported [79]. In the case of polydopamine, the presence of functional groups such as catechol, amine, and imine in its molecular structure makes it a prime candidate for use in biomedical applications [80] with the ability to self-organize into microstructured films [81].

Contrary to electropolymerization, electro-coupling relies on non-propagative covalent couplings induced electrochemically between functional nano-units bearing adequate moieties. To date, the alkyne-azide “click” cycloaddition *(i)* and the dimerization of 9-alkylcarbazoles *(ii)* are the most studied systems enabling the electrosynthesis of polymer and hybrid films [70,82].

*(i)* The alkyne-azide “click” cycloaddition, specifically leading to covalent triazole bonds under mild and aqueous conditions, is typically catalyzed by Cu(I) ions. The generation of these ions, which are unstable in water, by the electrochemical reduction of Cu(II) ions (using −0.3 V vs. Ag/AgCl), induces and confines the “click” coupling reaction in the vicinity of electrodes [70]. This film assembling strategy has been carried out with various weak polyelectrolytes (PAA, PAH) grafted with alkyne/azide groups, yielding covalently reticulated single-polyelectrolyte coatings that reversibly swell in response to post-assembly pH changes [70]. Accordingly, the covalent incorporation of fluorescent bis-pyrene moieties in such coatings composed of PAA evidenced the buffer behavior of the film, which maintained its internal pH at a value close to 3.5, while the environmental pH was changed from pH 4 to 7 [60]. In most recent examples, this click electrosynthesis approach has been combined with the electropolymerization of PANI [78] and adapted to hybrid nano-units, leading to pH-responsive nanostructured coatings with cargo encapsulation/release abilities (Figure 2C) [62].

*(ii)* Various organic and inorganic nano-units have been substituted in the 9- position of 9-alkylcarbazoles, enabling their covalent coupling by the electrochemical dimerization of carbazole moieties in acetonitrile [59]. This unique behavior was attributed to the selective activation of 3- and 6- positions of carbazoles and exhibited sensitivity to the potential applied: +1.0 V (vs. Ag/AgCl) favored more dimerization while +1.2 V also led to oligomerization [83]. The obtained organic or hybrid films benefited from internal layered structures and unlocking optical limiting applications [84,85]. Following this concept, recent efforts to potentialize other electrochemical dimerization reactions for functional films have been reported, opening promising perspectives for multifunctional systems [86].

**Table 3 molecules-27-03263-t003:** Typical electrodeposited systems based on weak polyelectrolytes and emerging applications.

Electrodeposition Principle	Polyelectrolyte Type and Typical Conditions (vs. Ag/AgCl)	Applications	Reference
Electrocoupling by click reaction	P AA or PAH grafted with alkyne and azide (−0.3 V, 0.6 mM CuSO_4_, H_2_O)	pH sensors, triggered release	[60,70]
Controlled dimerization	Dimerization of alkylcarbazoles (+1.0 V/+1.2 V, acetonitrile)	Photovoltaics	[87]
Electropolymerization	PANI (1.0 M HNO_3_ aqueous, 2 mA cm^−2^) PANI (0.5 M H_2_SO_4_ aqueous, CV −0.6 V/+1.5 V) Polycarbazoles (+1.3 V, aqueous or acetonitrile) Polydopamine (0.1 M phosphate buffer saline, CV −0.5 V/+0.5 V)	Capacitors, Capacitor sensors, Opto-electronic and electrochemical Sensors, biocoatings	[61,78] [76] [88]
Electrochemically induced precipitation	Chitin (+1.2 V with Fe^2+^ ions) CHI (+1.5 V with Cu(s)) CHI (+1 to +3 V) Collagen (pH 3.5; 0.1 M H_2_O_2_, 8 mA/cm^2^) ALG (oxidation of oxides, 1.7–4.4 mA/cm^2^) PAH (+0.6 V, with MoO_4_^2−^)	Drug release Sensor Drug delivery, biocoatings Biomaterials and actuators Wound treatment Implant coating	[89] [63] [90] [91] [69,71] [68]
Electrochemical co-deposition	HA + Polydopamine (+1 V in PBS buffer)	Antifouling	[92]
Complexation by pH-induced shift of PAH protonation	PAH/PAA (−0.5 V, 0.12 M H_2_O_2_) PAA/protected PAH and polyampholytes (H^+^ generation with 90 µA rate)	None reported	[65,66,67]

## 3. Layer-by-Layer Films and Vectors from Weak Polyelectrolytes

Over the past few decades, the Layer-by-Layer (LbL) strategy has enabled adsorbed polyelectrolyte films with unprecedented functional versatility. This approach, pioneered by Iler in 1966 and developed by Decher from 1991 [16], relies on the sequential adsorption of complementary chemical species (including polyelectrolytes), yielding coatings with controllable thickness whose growth can be driven by a range of interactions, including electrostatics, H-bonding, molecular recognition, and coordination (Figure 3) [93]. LbL films have thus attracted attention relating to their use in Nanoarchitectonics, owing to their potential for assembling functional nano-units on a large variety of substrates and topologies [94]. In this context, weak polyelectrolytes are mostly positioned as building blocks granting reversible interaction abilities to multilayers.

### 3.1. Layer by Layer Films

Weak polyelectrolytes of both synthetic and natural origin have been incorporated in LbL films, influencing their structure and composition [96], as well as their response to stimuli. Such films have thus been readily employed as biointerfaces and platforms with tunable mass transport properties [93] (as described in Table 4). Accordingly, adjusting the charge density and conformation of weak polyelectrolytes via pH and ionic parameters during (*i*) and after (*ii*) assembling multilayer films has been exploited:

*(i)* Decreasing the charge density of weak polyelectrolytes, either by decreasing their ionization degree of by increasing charge screening by salts, typically favors coiled conformations (Figure 1). In the case of LbL film buildups relying on electrostatically driven interactions, assembly conditions with smaller polyelectrolyte charge densities result in less crosslinks and more loops between the layers, leading to larger thickness increments at each adsorption step of weak polyelectrolytes compared to conditions where an extended conformation of the polyelectrolyte is favored (corresponding to higher charge density). Accordingly, the growth mechanism of a polydiallyldimethylammonium chloride (PDADMAC)/ALG multilayer changed from linear to exponential behavior when the assembly pH value was changed from pH 10 (where both polyelectrolytes are ionized) to pH 3 (where ALG is protonated) [97]. It follows that the multilayer thickness, chemical composition, mechanical properties, permeability, and adhesion are largely determined by pH and ionic parameters during the LbL deposition process. Therefore, the swelling of PAH/PAA and PLL/HA multilayers in aqueous media increases when the charge density of their weak polyelectrolytes decreases, a behavior attributed to the reduced ionic crosslink density and to loopy chain conformations [98,99]. The same phenomenon controls the hardness and elastic modulus of PAH/PAA multilayers [100], leading to lower values being obtained for PAH(pH 7.5)/PAA(pH 3.5) when both polyelectrolytes have a smaller charge density than for PAH(pH 6.5)/PAA(pH 6.5). Consequently, the assembly pH has been demonstrated to influence the drug transport mechanism in PAH/PAA multilayer films (Figure 3) [95]. In contrast, when the LbL film buildup relies on H-bonding, increasing the ionization rate of weak polyelectrolytes follows the opposite trend: for instance, the buildup of PAA/poly(vinylpyrrolidone) multilayers was hindered at pH values where PAA had a high charge density [93,101].

*(ii)* Post-assembly changes in pH and ionic parameters have been extensively studied as external stimuli to weak polyelectrolyte multilayers with respect to their response in thickness, porous structure, and encapsulation/release changes [93,102]. For LbL films relying on electrostatic interactions, decreasing the charge density of weak polyelectrolytes, either by post-assembly pH changes or by increasing the solution ionic strength, resulted in the swelling of the films [103] and eventually their dissolution. Accordingly, swelling coefficients of, respectively, up to 5-fold and 8-fold were found for PLL/HA multilayers subjected to ionic strength [104] and pH [105] stimuli. The swelling process was accompanied by structural changes in multilayer films, such as the emergence of holes in PLL/HA films when the NaCl concentration was changed from 0.15 M to 0.48 M [105] and the growth of pores in (PEI/PAH)/PAA films after the post-assembly pH was decreased to pH 2 [106]. The potential of such porous coatings was tested for designing slippery liquid-infused porous surfaces [106,107]. Concurrently, the modulation of the mass transport properties of weak polyelectrolyte multilayer films by post-assembly treatment has attracted much attention [108], leading to systems with tailored functions (Table 4), including ionic current rectification [109], enhanced Li^+^ conductivity [110], and filtration [111]. Conversely, not only the ionic strength but also the nature and valence of the salt used as the post-assembly stimulus were found to modulate the properties of weak polyelectrolyte multilayers: exposing a PAH/PAA multilayer film to various concentrations and type of metal ions enabled changing their pore sizes from 54 nm to 1.63 μm. This behavior was ascribed to phase separation in the film induced by metal-ion coordination with PAH [112], and was used to trap silver ions in the film to selectively detect methylmercaptan gas concentrations as low as 20 ppb [113].

### 3.2. Colloidal Systems Based on LbL Multilayers of Weak Polyelectrolytes

The LbL film deposition process has been adapted to colloidal substrates, providing an alternative to polymer self-assembly for designing nanocarriers, including hollow capsules (Figure 4) [121]. This method requires a sacrificial template (such as melamine formaldehyde, polystyrene, poly(methacrylic acid), silica, calcium carbonates, hydrogel microspheres), which needs to be removed by calcination or etching, and can be used to develop biocompatible systems with encapsulated drugs. Nevertheless, such nanocarriers remain very efficient when applied to catalysis [122] and energy storage [123] or as antioxidants [124]. Recent development has gradually evolved from prominently using synthetic strong polyelectrolytes to weaker ones, including using PAH, PEI, PLL, and poly(N-isopropylacrylamide) (PNIPAM) as polycations, as well as PMAA and PAA as polyanions. The corresponding colloidal systems benefited from the better control and fine tuning of intermolecular forces by pH modulation [125], ionic strength [126], and temperature [127]. Biosourced charged species such as dextran [128], CHI [129], bovine serum albumin [130], and DNA have also been used as functional units to encapsulate and deliver genetic cargo [128] and even co-deliver a drug [131].

With their high degree of functionality as well as their versatility, weak polyelectrolytes have been widely used to develop new tools for precision medicine in order to simultaneously address precision diagnosis and precision therapy. In this context, the use of polymer nanocarriers offers three topological regions which can allow functionalization: the inner cavity, the surrounding shell, and the external surface exposed to the microenvironment (Figure 4). Polyelectrolytes have consequently led to the development of nanocarriers for applications in imaging and for therapeutic purposes (Table 5). The on-demand release of cargo can be achieved by external or endogenous stimuli and provide a variety of ways to control the dosage, time, and location of release [132]. Temperature stands as a prominent stimulus, allowing changes in the hydration degree or layer organization which can be exploited for permeability changes and the release of the carrier [126]. Thermal responses can also be achieved by the use of inorganic nanoparticles such as gold [133], silver [134], and nanodiamond [135]. Under light irradiation, the local heating of nanoparticles can induce the rupture of the polymer shell or the modification of its permeability. This has been particularly demonstrated with polyelectrolytes such as PAH and poly(styrenesulfonate)(PSS) embedded with gold nanoparticles. This study demonstrated that, upon local heating by the gold nanoparticle, cargos retain their biological activities, although their diffusion within the cell is slightly decreased [136]. Ultrasounds can also be exploited as external stimuli to control the rupture of polyelectrolyte capsules. Ultrasounds are commonly used in imaging and ablative therapy. In the latter case, the mechanical deformation induced causes the bursting of the capsules, thus releasing their payload [137]. Magnetic fields have also been used to trigger the release of polymer capsules [132]. With the combination of magnetic nanoparticles such as iron oxide and electromagnetic fields of various frequency and power, different mechanisms can be induced. At low frequencies, a non-heating process can preferably be used to preserve tissues as well as a bioactive payload such as enzymes or DNA [138]. Meanwhile, with higher-frequency magnetic fields, a high increase in the local temperature induces the destruction of cellular and subcellular structures [139].

## 4. Gels and Vectors Based on Weak Polyelectrolytes Complexes

Processing weak polyelectrolyte into gels and vectors has been a very active research field for the past few decades. A range of approaches yielding prominently biosourced single-polyelectrolyte gels have been proposed based on precipitation/coagulation through H-bonding, hydrophobic interactions, and crosslinking through reactions with either ionic or covalent crosslinkers [152,153]. Although this research will not be covered in this review, recently published works are highlighted for further reading on this topic [154,155,156]. In contrast, processing polyelectrolyte complexes formed from the spontaneous entropy-driven complexation of polyanions and polycations [34] into gels and membranes is a method that has only been developed recently [13,157]. This section will therefore address the existing approaches for processing weak polyelectrolyte complexes into gels and membranes before describing selected systems that are relevant for pharmaceutical vectorization.

### 4.1. Gels Based on Weak Polyelectrolyte Complexes

Polyelectrolyte complexation by the electrostatic association of oppositely charged polymers results in phase separation from the solution through the formation of either solid precipitates or liquid complex coacervates. Although fundamental studies to elucidate the exact nature and behavior of these different complexes are still ongoing [9,11,12], many recent studies have focused on using polyelectrolyte complexes for applied materials, including vectors (described in Section 4.2) and gels. Adjusting the density of electrostatic crosslinks in polyelectrolyte complexes is critical to ensure their material processability. In that context, processable gels made of PDADMAC/PMAA complexes have been developed by screening polyelectrolytes’ charge densities with high-ionic-strength solutions (e.g., 2.5 M NaCl) followed by compaction through ultracentrifugation [14]. The obtained gels are named “saloplastics” or “compacted complexes of polyelectrolytes (COPEC)” and correspond to the blending of polyelectrolyte chains at the molecular level, where charges are reversibly compensated either intrinsically between polyelectrolytes or extrinsically with counterions (Figure 5). This gel elaboration approach has been successfully applied to several other weak polyelectrolytes, including PAA, ALG, PAH, and CHI, yielding self-healing gels with applications as biomaterials and catalyst supports (Table 6). The compaction process by ultracentrifugation initially represented a bottleneck for the larger-scale production of COPECs, triggering the development of alternative synthesis approaches based on simple centrifugation [158], injection [159], and sedimentation [160]. The properties of the resulting saloplastics vary greatly with the charge density and balance of their polyelectrolyte components both during and after synthesis, typically adjusted by pH and ionic force parameters. It follows that COPECs containing weak polyelectrolytes are dynamic stimuli-responsive materials that enable adjusting a large range of properties, including mechanical properties, composition, porosity, and sorption/release ability (Figure 5). Conversely, we reported on a dramatic increase in the porosity of PAH/PMAA COPECS following Na^+^ to Cu^2+^ cation exchange by complexation in the gel, enabling the in situ synthesis of catalytic nanoparticles [8]. The incorporation of biocompatible weak polyelectrolytes (e.g., CHI, ALG) and their cyclodextrin-grafted derivatives has attracted a great deal of attention in the field of biomaterial design [161,162,163].

Polyelectrolyte complexes have also been processed into functional membranes by aqueous phase separation [13]. Briefly, polyelectrolytes are first blended at the molecular level in an aqueous solution where the pH or ionic strength values do not allow their complexation, and the subsequent change in that parameter enables the precipitation of polyelectrolyte complex membranes [13]. This approach provides control over membrane pore size and structure in ways analogous to traditional non-solvent-induced phase separation. The synthesis pathway based on pH stimulus was developed with mixtures of weak and strong polyelectrolytes, yielding membranes made from PEI/poly(styrenesulfonate)(PSS), PAH/PSS, and PDADMAC/PAA systems for use in nanofiltration and micropollutant removal [13,165,166].

### 4.2. Weak Polyelectrolyte Complexes for Pharmaceutical Vectorization

The emergence of new diseases and strains of micro-organisms requires constant evolution and research in the field of drug delivery and nanomedicine. The most straightforward strategy adopted by the pharmaceutical industry has been the synthesis of new drugs capable of combating these pathologies. This, in turn, triggers the demand for appropriate drug carriers that can be effectively loaded with these drugs and protect them until they are administered and delivered. Although great progress has been made in the development of new drug carriers, including weak polyelectrolyte complex systems, in recent years [167,168,169], there are still systems such as water-insoluble drugs that these strategies fail to encapsulate efficiently [170].

Polyelectrolytes used for pharmaceutical research must meet the requirements of biocompatible polymer systems and be suitable for use as carriers of active substances. In this sense, the use of weak biosourced polyelectrolytes such as chitosan or charged chitosan derivatives such as glycol-chitosan or N-dodecylated chitosan as polycations and natural polysaccharides such as alginate, pectin, or carrageenan as polyanions has received much attention in the design of polyelectrolyte complexes for drug delivery due to their excellent bioavailability and biodegradability [171,172,173]. In addition to the difficulty of efficiently encapsulating actives with poor water solubility, other challenges in drug delivery include (i) the development of drug delivery systems that provide the sustained release of the drug within a desired therapeutic window to ensure efficacy; (ii) non-specificity, toxicity, and lack of localized administration strategies for certain treatments such as chemotherapeutics; (iii) scalability; and (iv) the development of harmonized regulatory guidelines for the manufacture of nanotechnology products that require contact with the human body (Figure 6). All the challenges cited above make the design of drug delivery systems much more complex than that of non-biological material release. As a result, the construction of efficient drug delivery carriers is usually achieved by assembling several components, each with its own role in the unified delivery function. An interesting way to achieve this is to use nanoarchitectonics approaches to develop biocompatible weak polyelectrolyte complexes formed in water with stimuli-responsive properties. The charges on weak polyelectrolytes are dynamic, causing polymer chains to adopt different equilibrium conformations even with relatively small changes to the surrounding environment [174] (Figure 1 and Figure 2). For instance, phosphonium polymer has been demonstrated to be able to control the physical and biological properties of sodium hyaluronate/phosphonium polyelectrolyte complexes [175]. The network swelling and therefore drug release rates of these systems can be controlled by varying the concentration of salt in the medium. Thus, while more hydrophilic molecules such as adenosine-5′-triphosphate can be released over 1–2 days, the sustained release of fluorescein and diclofenac over 60 days can be achieved, which is much longer than that previously reported for polyelectrolyte complexes [176,177,178]. On the other hand, only phosphonium polymers, including phenyl substituents, have shown a low cytotoxicity. Another example of improved control of drug release is in chitosan/alginate biocompatible pH-responsive polyelectrolyte complexes, which were developed as less invasive delivery systems for oral insulin administration [179]. This association of polyelectrolytes allows the delivery system to withstand prolonged contact with acidic gastric media and enzymes in the gastrointestinal tract, enhancing bioavailability by controlling insulin release in the intestinal tract. Furthermore, the examined polyelectrolyte complexes exhibit non-cytotoxicity against Caco2 cells.

Nanoarchitectonics approaches to develop drug reservoirs with collective nanosystem functionality have been used to address the clinical limitations of premature drug release and tumor non-specificity. For instance, a novel superparamagnetic chitosan-based nanometer-sized colloidal polyelectrolyte complex integrating the water-soluble polymeric prodrug poly(L-glutamic acid)-SN-38 (PGA-SN-38) was designed using a one-shot manufacturing process to efficiently deliver SN-38 [180]. The combination of these systems enhanced drug solubility and tumor-targeting accumulation, thereby improving the therapeutic efficacy against colorectal cancer in vivo (tumor suppression rates of up to 81%). Interestingly, although the prepared material exhibited controlled release at pH 7.4, a burst release of the drug was observed during the first 12 h, which was attributed to the dissociation of the PGA-SN-38 prodrug from the nanopolyelectrolyte complexes due to the partial instability of the chitosan-based nanocomplexes in phosphate-buffered saline medium. The standardization and scale-up of polyelectrolyte complexes obtained through bottom-up methodologies is still a great challenge and necessitates carrying out arduous experimentation, since it depends on multiple intrinsic and extrinsic variables [181]. A means to improve the main problems generated during polyelectrolytic complexation, such as obtaining large particle sizes and highly polydisperse systems, is to employ top-down methods instead [182]. However, top-down methods such as high- or ultra-high-pressure homogenization also have some disadvantages, such as the chemical degradation of the material by excessive energy applied during the disaggregation process [183]. Nevertheless, the great benefit of these techniques is that the conditions implemented can be easily reproduced and scaled-up to industrial level. Thus, the polyelectrolyte complexes developed under these methodologies are suitable for easy technology transfer. Finally, the opportunities offered by nanotechnology in the health sector are also accompanied by challenges in the regulation of these products. One example of these concerns is the modification of the physicochemical properties of nanomaterials, which can lead to altered toxicity, solubility, and bioavailability profiles [184,185,186]. In addition, evidence regarding the potential safety issues of synthetic polymers appears to be the main driver of research on the use of natural polysaccharides in the application of more recent responsive polyelectrolyte complexes as drug delivery systems. Furthermore, the existence of strong regional differences in the regulation of nanomedicines confirms the need for the harmonization of information requirements for nanospecific properties. Current efforts are directed towards gaining sufficient knowledge on the quality, safety, and efficacy of nanomaterials to support regulatory decisions and enable a smooth transition to clinical applications (Table 7).

## 5. Block Copolymer Systems Based on Weak Polyelectrolytes

Block copolymers (BCPs) based on weak polyelectrolytes combine a stimuli-responsive nature with native self-assembling properties, positioning them as an interesting class of polymeric nano-units for use in structured materials. The incorporation of weak polyion blocks in BCP facilitates copolymer synthesis and characterization while leading to weak bonding interactions *(i)* that facilitate the reversible conformation of the self-assembled nanostructures with reversible complexation abilities *(ii)* towards drugs, metallic ions, and other types of cargo. Since the charge density on weak polyelectrolyte is pH-dependent, it also enables the dissolution of hydrophobic/ionic BCPs in polar organic solvents [204,205]. Emerging applications include agents that can be used to stabilize and/or deliver dugs, peptides, and small molecules; lubricants; colloids; the patterning of nanostructures; the synthesis of stimuli-responsive capsules; and filtration membranes [206,207,208,209]. Chemically, BCPs are formed by the covalent linkage of two or more distinct monomer units grouped in discrete blocks along the polymeric chain [210]. Due to the rapid progress made in polymer synthetic strategies and techniques such as controlled polymerization along with facile post-polymerization functionalization, BCPs with well-defined molecular weights and macromolecular architectures can be synthesized [211]. The exceptional compositional and molecular structural versatility of BCPs has facilitated the tremendous growth and application of new synthetic routes, enabling high levels of architectural complexity to be achieved. Examples of weak PE-based BCPs are given in Table 8. This section will highlight the scope of BCP-based PEs and their applications.

### 5.1. Directed Self-Assembly of BCPs for Nanopatterning

BCP self-assembly is a versatile process where the incompatible behavior of the constituent blocks leads to the formation of microstructures with a plethora of potential applications [211,212,213]. In this context, the directed self-assembly of BCPs has revolutionized the modern nanoelectronics industry. It has facilitated long-ordered structures, the transfer of patterns, and the design of low-resolution high-density nanostructures for high-level computing applications (Figure 7). There are some major drawbacks of current nanofabrication techniques, such as photolithography (limited in feature size), electron-beam lithography (low throughput), and EUV lithography (high development costs) [210]. Owing to these challenges, a large volume of research is dedicated to the development of cost-effective nanofabrication technology. In this case, the BCP soft lithography method offers a very simple, scalable, cost-effective platform for use in nanoscale fabrication, where feature sizes and geometries can be tuned via the chain length and volume fraction of BCPs. BCPs can self-assemble into varied different nanostructures through microphase separation, which is driven by the enthalpy of the demixing of the constituent components of the BCPs, while the process is constrained by the chemical connectivity of the blocks [214]. BCPs can be tuned to self-assemble into desired nanostructured morphologies such as spheres, cylinders, lamellae, and gyroids by adjusting the volume fraction (*f*) of the constituent blocks [212,214]. In addition, chemical heterogeneity between the BCP blocks allows the selective complexation of guest species (e.g., metal ions), thereby creating patterned etch masks that can be transferred into functional materials and surfaces (Figure 7) [213,215]. Accordingly, systems based on BCP containing weak polyelectrolyte blocks such as poly(styrene-*b*-4vinylpyridene) (PS-*b*-P4VP) [216] and PS-*b*-P2VP [217] can be used to produce well-defined nanostructures with long-range ordered morphologies. Upon the infiltration of weak polyion block domains with metal ions (Ni^+^, Al^3+^ and Cr^2+^), well-defined metal-oxide nanodots and/or nanowires were obtained and used to define etch masks for pattern transfer [217]. For instance, we recently reported the use of a high-contrast etch mask for pattern transfer into silicon substrates through ICP/RIE etching for the development of vertically coupled plasmonic arrays for use in surface enhanced Raman scattering [213], anti-reflective surfaces, and photocatalysts [218].

### 5.2. Colloidal Systems from Weak Polyelectrolytes BCP for Drug Delivery

BCP-containing polyelectrolytes blocks have been used to design nanometer- to micrometer-sized vectors for use in drug delivery and encapsulation applications through manipulating the physicochemical properties of shells to adjust their permeability (Figure 8) [220]. Stimuli such as pH and ionic strength changes create the opportunity to easily control drug-loaded vectors in open and closed states [221,222,223].

Weak polyelectrolyte-based BCPs with constituent blocks such as carboxylic acids, phosphoric acid, and amines show different ionization degrees depending on the pH of the surrounding environment [225]. This leads to conformational changes due to different interactions among polymeric chains and also to the formation of hydrogen bonds. The solubility conditions and relative size of the blocks determine whether they will self-assemble into micelles, vesicles, or a combination of both [226]. When BCPs combine weak polyion blocks with hydrophobic blocks (e.g., PS-*b*-P2VP and poly(styrene-b-acrylic acid) (PS-*b*-PAA)), they form core-shell micelles in water with a hydrophobic core and polyelectrolyte corona [224]. Such systems enable solubilization of hydrophobic molecules into their core, while the external corona serves stabilizes vectors in aqueous media and helps binding to oppositely charged molecules, e.g., proteins. Conversely, changing the surface charge density of such micelles by pH-induced protonation of weak polyelectrolyte blocks controlled the stability or aggregation behavior of PS-*b*-P2VP-*b*-PEO [227], PB-*b*-P2VP-*b*-PMAA [228], and PCL-*b*-PEO-*b*-P2VP)[229] micelles. In contrast, when BCPs combine weak polyion blocks with hydrophobic blocks, they can micellize on complexation of their polyelectrolyte block with oppositely charged partners (polycations, drugs, multivalent cations) as reported for poly(ethylene glycol-b-2-(dimethylamino)ethylmethacrylate) (PEG-*b*-PDMAEMA), poly(acrylic acid-*b*-ethylene oxide) (PAA-*b*-PEO) and PNIPAM-*b*-PAA [230,231]. Such micelles recently found applications as drug vectors and sol-gel structure directing agents with pH-triggered stability [232].

### 5.3. Membranes from Weak Polyelectrolytes BCP for Filtration

Global population growth, climate change and human activities have degraded the quality of fresh water to critical levels through the discharge of chemicals into ground and surface waters. For filtration BCP based membrane technology has gained great significance utilizing nanoscale features as the pores of the membrane granting hydraulic permeability and separation selectivity. The self-assembled nanostructure of BCPs in the equilibrium state offer their (1) utility in the fabrication of high-performance separation membranes, (2) limitless chemical compositions, and (3) scalable nanomanufacturing processes. Manipulation of the nanostructure of BCP membranes provides wide range of limits of size-selection, ability to tailor the surface chemistry of BCP membranes for different water reuse. BCP membranes rely on the swelling of blocks of copolymer to narrow the pore size into the nanofiltration regime. In that context, the response of weak polyelectrolyte blocks (e.g., PAA, P2VP) to salt and pH stimuli via conformational changes that changes the pore geometry is critical (Figure 1 and Figure 9) [207].

BCP membranes in the nanofiltration regime have also been obtained through the blending of different BCPs such A-B/A-C type blocks. For example, the combination of PS-P4VP and PS-PAA BCPs allows the formation of smaller pore sizes due to the hydrogen bonding occurring between PAA and P4VP blocks of the two blends. These complexes drive the morphological shift towards densely packed spherical nanostructures that form pores with smaller dimensions [233]. Another relatively easy process used to fabricate membranes is non-solvent-induced phase separation (NIPS) [234], where a concentrated BCP solution is casted on a polished surface and subsequently immersed in a non-solvent, leading to phase separation. In the case of BCPs, this process results in solvent evaporation and microphase separation, yielding unsymmetrical membrane where a mesoporous skin layer is supported by a macroporous bulk structure. NIPS has been employed in membrane preparation from PS-*b*-P4VP [235] and polystyrene-block-poly(N,N-dimethylaminoethyl methacrylate) (PS-b-PDMAEMA) BCPs [236]. In the latter example, the PDMAEMA block in the membrane is pH- and temperature-responsive, modulating the pore size in the range of 20–80 nm because of swelling/de-swelling. In PS-*b*-P4VP-based membranes, the P4VP segment is deprotonated at higher pH values and collapsed onto the pore walls, yielding a higher water permeability and larger pore sizes. Moreover, in PS-*b*-P4VP, the complexation of the P4VP segment with metal ions in the casting solution provides control over the pore formation [237].

**Table 8 molecules-27-03263-t008:** Recently reported BCPs containing a weak polyelectrolyte block and their applications.

BCP Polyelectrolyte	Chemical Structure	Applications	References
Poly(styrene)-b-poly(4-vinylpyridine)PS-*b*-P4VP	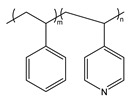	Pattern transfer filtrationcolloids	[210,238]
Poly(styrene)-b-poly(2-vinylpyridine)PS-*b*-P2VP	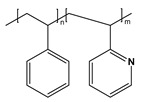	Etch masks, drug delivery filtration	[213,215,216]
Poly(styrene)-b-poly(acrylic acid)PS-*b*-PAA	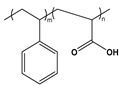	Ultra-filtrationDrug deliveryNP synthesis	[233,239]
Poly(styrene)-b-poly(methacrylic acid)PS-*b*-PMAA	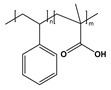	FiltrationPattern transferdrug delivery	[240]
Poly(ethylene oxide)-b-poly(2-vinylpyridine)PEO-*b*-P2VP	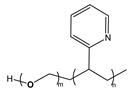	Membranes	[241]
Poly(ethylene glycol)-b-poly(2-(dimethylamino)ethyl methacrylate)PEG-*b*- PDMAEMA	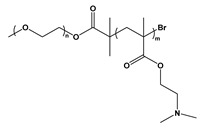	Drug delivery	[242]
Poly(ethylene oxide)-b-poly(acrylic acid)PEO-*b*-PAA	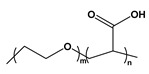	Drug delivery vehicle	[243]
Poly(ethylene oxide)-b-poly(acrylic acid)-b-poly(styrene)PEO-b-PAA-b-PS	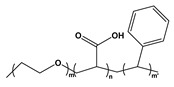	Controlled drug delivery	[244]
Poly(n-butyl acrylate)-b-poly(acrylic acid)P*n*BA-*b*-PAA	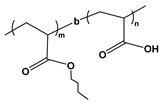	Nanoreactors NP synthesis	[245]
Poly(styrene)-b-poly(L-lysine)PS-*b*-PLL	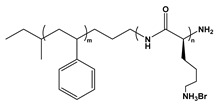	DNA carrierEncapsulation	[246]
Poly(N-isopropylacrylamide)-b- poly(acrylic acid)PNIPAM-*b*-PAA	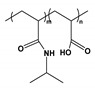	EncapsulationDrug delivery	[247]

## 6. Closing Remarks

From surface platforms to colloids and gels, weak polyelectrolytes have enabled the development of a wide range of functional materials owing to their intrinsic response to stimuli, including solvents, temperature, pH, and salt. This review paper is focused on the latter two stimuli, as these simple parameters are able to induce several responses in weak polyelectrolyte chains at the molecular level, including changes in their ionization rate, charge density, H-bonding ability, and conformation. In turn, these changes are transmitted to the supra-molecular level, enabling nano-systems to respond in precisely predetermined manners, including by changing their stability, their structure, and their physical properties (adhesion, solubility, physisorption, mechanical properties, etc.). Furthermore, the chemical variety of weak polyelectrolytes, encompassing natural and synthetic polymers with a range of available chemical functions and ionization constants (pKa and pKaH), enables the creation of varied biocompatible materials and vectors that hold promise for use in future biomedicine and pharmaceutical applications. Given this background and considering that weak polyelectrolytes can rely on external stimuli as both assembly and post-assembly triggers, it is not surprising that they stand as promising tools for the nanoarchitectonics of organic and hybrid systems. This dual role enables weak polyelectrolytes to act not only as functional nano-units conferring “smart” stimuli-responsiveness to materials but also as structure-directing agents. In this respect, prominent future developments can be anticipated in at least two research directions. First, pharmaceutically relevant systems based on biocompatible weak polyelectrolytes will gain improved precision and (multi)functionality for target drug delivery. Concurrently, nanostructured systems, including those based on BCP, will enable the confinement of weak polyacid/polybase nanodomains and pores to support the emergence of next-generation membranes and electrodes for use in energy and environment sustainability devices.

## Data Availability

Not applicable.

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
