# Peer review of "Weak Polyelectrolytes as Nanoarchitectonic Design Tools for Functional Materials: A Review of Recent Achievements"

_molecules, 2022, doi:10.3390/molecules27103263_

Round 1
Reviewer 1 Report
Comments from Reviewer
Title: Weak polyelectrolytes as nanoarchitectonic design tools for functional materials
The current form's presentation of methods and scientific results is satisfactory for publication in the Molecules journal. The minor and significant drawbacks to be addressed can be specified as follows:
1. Lines 33 and 34. The use of first names and/or name initials is not recommended.
2. Line 48, “stimuli. .” Two dots!
3. Fig. 1. Mw? Mass weight? I suggest Mw -ïƒ Mass weight.
4. Line 166, Dahlin et al. suggested. But [49] Ferrand-Drake del Castillo, (…). Dahlin et al. ---> Ferrand-Drake del Castillo et al.
5. I would suggest adding a list of abbreviations.
6. Line 202. table 3 or Table 3. Standardize.
7. Tab. 3. 0.6v? or 0.6V?
8. Fig. 3(B). The part in the middle is not very expressive.
9. Tab. 4. (i) Ionic Current Rectification ---> Ionic current rectification (ii) Chondroitin Sulfate ---> Chondroitin sulfate
10. Fig. 4. Deposition ---> Deposition.
11. Line 371. PSS?
12. Tab. 5. (i) Chemical Structure ---> Chemical structure. (ii) [141] [142] ---> [141,142]. See other tables.
13. Tabs. 7 and 8. The same problems as in case of Table 5.
14. Line 562. “.[211]” ---> “[211].”
15. Figs. 8 and 9. Permissions?
16. Fig. 9. DIB? Mel?
17. I like this article, but there are still a lot of minor errors/typos. I realize that this is a flaw in review papers. I also suggest emphasizing in the title that this is a review.
Sincerely,
The reviewer.
Author Response
We thank the reviewer for his comments.
We provide a point-by-point answer file addressing the issues raised by the reviewer

Reviewer 2 Report
The manuscript entitled “Weak polyelectrolytes as nanoarchitectonic design tools for functional materials” tried to summarize the recent 5 years of progress in nanoarchitectonic systems formed by weak polyelectrolytes and their applications. A large amount of literature was reviewed. However, the manuscript suffers logical issues. I recommend major revision before considering publication.
(1) Introduction, Page 3
The hierarchy of the manuscript is rather confusing. For example, “In the initial part, organic and hybrid thin films of weak polyelectrolytes whose cohesion relies on covalent, hydrophobic and H-bonding interactions are described.” “The following sections describe reversible systems based on polyelectrolyte complexes formed from spontaneous entropy-driven complexation of weak polyanions and polycations.” It is hard to understand how the authors categorized the research work.
The authors stated that “The final part addresses the growing significance of block copolymers (BCP) containing weak polyelectrolyte blocks for nanostructuring surfaces, colloids and membranes.” However, in the first part, some work about the synthesis and usage of BCP were discussed. The authors need to revise the discussion to make it coherent.
(2) Page 3, line 78-81, line 88-89
The authors claimed in line 78-81 that “Based on these backgrounds, weak polyelectrolytes offer a range of non-covalent interactions including electrostatic, H-bonding and hydrophobic interactions which can not only tailor the properties of polymeric matrices, films and colloids but also direct their 80 structuration at nano and micron levels (figure 1).” The readers are expected that the discussion would not include interactions based on covalent bond. However, in line 88-89, “In the initial part, organic and hybrid thin films of weak polyelectrolytes whose cohesion relies on covalent, hydrophobic and H-bonding interactions are described.” The authors need to revise the structure of the manuscript.
(3) Table 1,
In Table 1, the authors listed most common weak polyelectrolytes and their pKa/pKaH. It is advised to add a column to list the molecular weight and environment since they affect these values of polyelectrolyte.
(4) Most tables are not informative. The full names of block of copolymers should be given and then the abbreviation can be used. Some tables contained information that other tables have listed. In Table 8, please correct the chemical structure of PEO-b-P2VP.
(5) Figure 9,
What are “DIB” and “Mel”?
(6) Some sentences need to be revised, for example:
Page1, line 42-43
Page 2, line 47
Page 3, line 86
(7) There is no need to capitalize the polymerization strategy, such as Nitroxide-Mediated Polymerization, AtomTransfer-Radical-Polymerization.
Author Response
We thank the reviewer for his constructive critics.
Please find our answers in the file enclosed to this email.
